# Development and Improvement of a Piezoelectrically Driven Miniature Robot

**DOI:** 10.3390/biomimetics9040226

**Published:** 2024-04-09

**Authors:** Guangping Wu, Ziyang Wang, Yuting Wu, Jiaxin Zhao, Feng Cui, Yichen Zhang, Wenyuan Chen

**Affiliations:** 1National Key Laboratory of Advanced Micro and Nano Manufacture Technology, Shanghai Jiao Tong University, Shanghai 200240, China; excalibur996@sjtu.edu.cn (G.W.); wangziyang@sjtu.edu.cn (Z.W.); yutingwu@sjtu.edu.cn (Y.W.); jiaxin.zhao@sjtu.edu.cn (J.Z.); zhangyic@sjtu.edu.cn (Y.Z.); 2Department of Micro/Nano Electronics, School of Electronic Information and Electrical Engineering, Shanghai Jiao Tong University, Shanghai 200240, China; chenwy@sjtu.edu.cn

**Keywords:** miniature robot, piezoelectric actuator, optimal design, frequency response

## Abstract

In this paper, we proposed a miniature quadrupedal piezoelectric robot with a mass of 1.8 g and a body length of 4.6 cm. The robot adopts a novel spatial parallel mechanism as its transmission. Each leg of the robot has two degrees of freedom (DOFs): swing and lift. The trajectory necessary for walking is achieved by the appropriate phasing of these two DOFs. A new manufacturing method for piezoelectric actuators was developed. During the stacking process, discrete patterned PZT pieces are used to avoid dielectric failure caused by laser cutting. Copper-clad FR-4 is used as the solder pad instead of copper foil, making the connection between the pad and the actuator more reliable. The lift powertrain of the robot was modeled and the link length of the powertrain was optimized based on the model. The maximum output force of each leg can reach 26 mN under optimized design parameters, which is 1.38 times the required force for successful walking. The frequency response of the powertrain was measured and fitted to the second-order system, which enabled increased leg amplitudes near the powertrain resonance of approximately 70 Hz with adjusted drive signals. The maximum speed of the robot without load reached 48.66 cm/s (10.58 body lengths per second) and the payload capacity can reach 5.5 g (3.05 times its mass) near the powertrain resonance.

## 1. Introduction

Miniature robots generally refer to robots with feature sizes less than 10 cm [1]. In recent years, robots at this scale have become a hotspot for researchers. In particular, due to their small size, light weight, and agility of movement, miniature robots will play an irreplaceable role in search and rescue, exploration, medical diagnostics, and fault detection in confined environments [2,3,4,5]. With the development of new materials and processes, the drive elements of miniature robots are also enriched. These elements include electromagnetic motors [6,7], shape memory alloys (SMAs) [8,9], dielectric elastomers [10,11], and piezoelectric ceramics [12,13,14,15,16]. Piezoelectric elements are widely used as actuators for miniature robots because they have the advantages of a fast response, a high power density, and no electromagnetic interference.

The payload capacity and speed of the miniature robots are directly related to the motion performance. The greater the load capacity, the more sensors the robot can carry, and, thus, the more functions and tasks the robot can perform; the faster the speed, the shorter the time it takes for the robot to reach its destination. Consequently, it is necessary to increase the payload capacity and speed of the miniature robots. There are several works of research on the kinematics model of miniature robots [17,18,19], which are mainly used to guide the gait design and trajectory planning for robots. In response to the increased performance of the Harvard Ambulatory Micro-Robot (HAMR) series of robots, the actuation model of the robots was developed, leading to the improved design of the actuator and transmission [20,21]. The output performance of the actuator also affects the motion performance of the robots, so the exploration and optimization of the actuator manufacturing process is also the focus of the research. Wood et al. not only studied various manufacturing processes for piezoelectric actuators [22,23], but also optimized the morphology parameters of the actuators, resulting in a significant increase in energy density compared to existing commercial piezoelectric bending actuators [24]. In addition, a non-linear, dynamic model of the flexure-based transmission in the HAMR was presented and the transmission was redesigned based on experimental data to improve the performance in the quasi-static mode [25]. To take full advantage of the excellent performance of the actuator in resonance, researchers at the University of Newcastle have developed a six-legged piezoelectric robot that can reach speeds of up to 55 cm/s [26,27,28].

In this paper, we designed and manufactured a miniature piezoelectric quadrupedal robot named SMR-O, which includes exoskeletons, powertrains, four legs, and eight piezoelectric actuators. We designed a new spatial parallel mechanism that mimics the motion joints of insects, which can achieve two degrees of freedom of motion. The leg is connected to this motion joint, and each leg has two output degrees of freedom (DOFs): swing and lift. The robot’s gait is a common trot gait of insects, which is realized by controlling the phase of the drive signals of eight actuators. The robot can move forward and turn by appropriate phase inputs to the eight piezoelectric actuators. Compared to the previous version of the robot SMR-M designed by us [29], which mainly focused on the design of the transmission mechanism and exploration of the robot manufacturing process, this paper conducted a systematic analysis and optimization of the robot powertrain and improved the fabrication process of the actuator. We proposed a novel process for the batch manufacturing of piezoelectric bending actuators, which allows for a more reliable bonding between copper traces and actuators and avoids the shorting of the PZT. Meanwhile, due to the use of alumina as the base and carbon fiber as the bridge at the PZT–alumina interface, the output performance of the actuator has also been significantly improved. Moreover, a drive model for the lift DOF of the transmission was developed, based on which the linkage dimensions in the transmission were optimally designed, enabling the robot leg to have a greater output force. The output performance of the actuators and the output force of the legs were measured. Furthermore, we experimentally obtained the frequency response of each powertrain and actuated the powertrain in the resonance regime to improve the motion performance.

The contributions of this paper include the following: (i) A manufacturing method for piezoelectric actuators different from HAMR has been proposed. During the stacking process, discrete pre-patterned PZT pieces are used instead of a whole PZT piece to avoid PZT dielectric failure caused by unexpected laser cutting during actuator release. Copper-clad FR-4 instead of copper foil is adopted as the solder pad to improve the reliability of the bonding between the pad and the actuator. (ii) The link lengths of the robot’s transmission mechanism and the robot’s leg lengths have been optimized based on the actuation model of the powertrain, which enables the robot legs to have a greater force to carry the robot. (iii) Because the motion of each robot leg is determined by the powertrain including the actuator and transmission mechanism, it is necessary to drive the robot’s powertrain instead of the actuator to a resonant state. We obtained the resonance frequency of the powertrain and its phase shift relative to the input signal through experiments and fitting. We drive the powertrain to a resonance state to make the robot leg produce a greater displacement and force; the motion performance of the robot has been greatly improved, with an unloaded speed of 48.66 cm/s and a payload capacity of 5.5 g. This locomotion performance is greatly improved compared with the speed of 5.32 cm/s and the payload capacity of 2.5 g at a drive frequency of 10 Hz.

The paper is organized as follows: Section 2 introduces the components of the robot and provides a detailed description of the hip joint of the robot as a transmission mechanism. In Section 3, a new manufacturing method for piezoelectric actuators is proposed. The composition of the actuator and the three steps of the manufacturing process are described in detail. In Section 4, the robot powertrain is modeled. Based on this model, the link lengths of the transmission mechanism and the robot’s leg lengths are optimized with the optimization objective of maximizing the leg output force. In Section 5, the output indicators of the actuator, the output force of the robot legs, the frequency response of the powertrain, and the locomotion performance of the robot are tested. Finally, in Section 6, the work of this paper is summarized.

## 2. Overall Design

The miniature quadruped piezoelectric robot uses eight piezoelectric actuators as its power sources and four identical spatial parallel mechanisms as its four hip joints; the hip joints form the robot’s transmission. The lift and swing inputs of the hip joint are fixed to the lift actuator and swing actuator, respectively. Flexure hinges based on polyimide film are used as rotating pairs in the parallel mechanisms. Each leg is driven by two actuators and the corresponding hip joint converts the reciprocating oscillation of the two actuators into the lift and swing of the leg. Eight actuators are soldered to the upper and lower circuit boards, which are affixed to the exoskeleton. The circuit boards introduce electrical signals to the actuator via copper traces. A CAD model of the designed miniature quadruped robot is shown in Figure 1a. The miniature robot has a body length of 4.6 cm and a mass of 1.8 g. Figure 1c shows the prototype of the manufactured miniature robot contrasted with a coin.

The hip joints are important components for motion translation and power transfer in robots. For the legs to produce the closed trajectories necessary for robotic motion, a novel spatial parallel mechanism is proposed as the hip joint, which consists of two structurally interconnected and kinematically decoupled slider-crank linkages. In other words, the lift and swing of the robot’s leg are independent of each other. The schematic diagram of the powertrain with a hip joint and two actuators is shown in Figure 1b.

For piezoelectric actuators with a small bending displacement, the movement of the end can be equivalent to linear movement, so it is reasonable to use the actuator as the input of the prismatic pair. The red dashed lines in Figure 1b represent the central axis of the flexure hinges. The first slider-crank linkage responsible for the robot’s lift motion consists of flexure hinges 1, 2, and 3 and links connected to these hinges. Similarly, the second slider-crank linkage for the robot’s swing motion consists of flexure hinges 4, 5, and 6 and links connected to them. The motion principle, kinematic analysis, and design methodology of the transmission mechanism are described in detail in [29].

## 3. Piezoelectric Actuator Manufacturing

There are two main differences between the manufacturing method of the actuator in this paper and the manufacturing method in [22]. One is the use of copper-clad FR-4 rather than copper foil as the pads because the copper foil can easily fall off during high-temperature soldering. The other is the use of discrete patterned pieces of PZT rather than a whole PZT piece to prevent the shorting of the PZT caused by unexpected high-power laser cutting during the actuator release process.

The core elements of the piezoelectric bimorph bending actuator are the carbon fiber (CF) layer located in the central layer and the two PZT layers above and below it. The prepreg CF is used to bond the two layers of PZT and to introduce the external electrical signals, as well as to provide a certain degree of stiffness for the actuator. The PZT is the active deformation layer that provides the prime force for the bending of the actuator. Considering that PZT is a brittle material, a strong base is necessary for a reliable connection of the actuator to the circuit board. At the same time, a rigid extension is necessary in order to withstand loads and amplify output displacements. Naturally, parasitic bending occurs at the interface between the PZT and the base (or the extension). To eliminate bending at the interface, rigid attachments such as FR-4 and carbon fiber are needed. Carbon fiber has a higher strength and is, therefore, more effective in preventing parasitic bending. However, in order to solder the actuator to the circuit board, additional copper foils acting as pads need to be bonded to the carbon fiber [22]. Typically, the bonding of the copper foil to the carbon fiber is not strong at high temperatures, and precise cuts that do not damage the carbon fiber are required to achieve the patterning of the copper foil. To face the problem, we use commercial copper-clad FR-4 instead of copper foil as the pads.

The piezoelectric actuators in this paper consist of PZT-5H, prepreg CF (including the central CF and CF attachment), alumina, prepreg glass fiber (GF), and copper-clad FR-4, as shown in Figure 2a. The central carbon fiber is sandwiched between two layers of PZT. Alumina in the same plane as PZT is used as the rigid base and the extension of the actuator. The prepreg glass fiber firmly adheres FR-4 to the actuator. The CF ‘bridges’ the PZT–alumina interface and provides a strong attachment preventing bending at this interface. Moreover, the patterned copper-clad FR-4 pads provide electrical connections between the PZT pieces and off-board power electronics. Three pads are connected to the bias voltage, the drive voltage, and the ground, respectively. The via hole is formed through the alumina and the glass fiber to make contact with the central carbon fiber as shown in Figure 2b.

The batch manufacturing process for actuators is divided into three main steps. The first step is to cut the required materials using the diode-pumped solid-state (DPSS) UV laser to obtain various components that need to be stacked, including individual pieces of PZT, alumina, central CF, CF attachment, glass fiber, FR-4 jig, and copper-clad FR-4. It is worth noting that we cut all of the way through the PZT at sufficiently low power to improve the flexure strength as much as possible. And the low laser power is to avoid the dielectric breakdown of the PZT. The main role of the prepreg glass fiber is to bond the FR-4 to the actuator by the resin in it, so a very thin glass fiber layer (30 μm in thickness) is selected to avoid adding extra mass to the actuator.

The next step is to stack the prepared components mentioned above. The stacking order of these components is shown in Figure 3a. The individual pieces of PZT and two alumina strips are placed carefully into the FR-4 jig, using instant adhesive to hold them in place. That is, the PZT, alumina, and FR4 jig are in the same layer and their positional relationship is shown in Figure 3b. In order to improve the dielectric strength of the PZT, cyanoacrylate (CA) glue is applied at the interface between the PZT and alumina and the edges of the PZT. The materials of other layers are stacked in order. The copper-clad FR-4 strip is on the same layer as the top CF and is firmly bonded to the actuator through the glass fiber, forming a solder pad for the actuator. The final stack is cured at a temperature of 120 °C to achieve the complete curing of the resin in carbon fiber and glass fiber. After curing is complete, a laminate with seven layers of materials firmly bonded together is formed; the top view schematic diagram of the stacked laminate is shown in Figure 3c.

The final step is to release each actuator from the laminate by laser cutting. Conductive epoxy is applied to the via hole and the nearby pad to form an electrical connection. It should be pointed out that there is a certain gap between the pad at the via hole and the CF attachment as can be seen in Figure 2b, which is to prevent an electrical connection between the two pads located on the same side of the actuator by the carbon fiber. A photo of the fabricated actuators with a length of 16 mm is shown in Figure 2c.

Figure 4a,b are stacked laminates with discrete patterned pieces of PZT and a whole piece of PZT, respectively. The black solid lines are the cutting paths of the laser. To improve the cutting speed, the high-power laser is used during the actuator release process. As shown in Figure 4a, when using discrete PZT pieces for stacking, the damage to the PZT by the high-power laser can be avoided by making the cutting path deviate from the PZT profile. However, when using a whole piece of PZT, to prevent laser damage to the PZT, it is necessary to cut the PTZ with a low-power laser and cut other materials with a high-power laser, and the laminate needs to be flipped for repeated cutting [22]. The requirements for cutting parameters are strict, and the process is complex.

## 4. Transmission Parameters Design

The parameter design of the transmission is crucial for miniature robots, affecting the transmission efficiency, as well as the robot’s payload capacity and speed of movement. In this section, we model and analyze the transmission system of the robot and optimally design its parameters.

The piezoelectric actuator is modeled as a force source *F_a_* in parallel with a spring *k_a_* and a damper *b_a_* and has a certain mass *m_a_* to describe the dynamic performance [30]. Since the lift and swing DOFs of the transmission are fully decoupled, the lift DOF including a crank-slider mechanism can be analyzed separately. The flexure hinge in the transmission is equivalent to a pseudo-rigid body model that includes a rigid pin joint and a torsional spring [20]. The illustration of the lift powertrain model is shown in Figure 5.

Assuming that the relevant parameters of the actuator and the displacement of the robot leg are known, the equation is obtained based on the kinetic energy balance:(1)$$\begin{array}{l} F_{a}\delta_{a}-\frac{1}{2}k_{a}\delta_{a}^{2}-b_{a}{\dot{\delta }}_{a}-\frac{1}{2}k_{1}\phi_{1}^{2}-\frac{1}{2}k_{2}\phi_{2}^{2}-\frac{1}{2}k_{3}\phi_{3}^{2}-F_{leg}\delta_{leg-y}=\frac{1}{2}m_{a}\Delta v_{a}^{2}+\frac{1}{2}m_{0}\Delta v_{0}^{2}\\ +\frac{1}{2}m_{1}\Delta v_{1}^{2}+\frac{1}{2}J_{1}\Delta \omega_{1}^{2}+\frac{1}{2}m_{2}\Delta v_{2}^{2}+\frac{1}{2}J_{2}\Delta \omega_{2}^{2}+\frac{1}{2}m_{leg}\Delta v_{leg}^{2}+\frac{1}{2}J_{leg}\Delta \omega_{leg}^{2} \end{array}$$
where *F*_a_ is the actuator block force generated when the end displacement is 0, which is a constant value under a determined drive voltage; *δ_a_* is the bending displacement of the actuator, which is variable; *k_a_* is the actuator stiffness; *b_a_* is the actuator damping; *k_i_* and *ϕ_i_* (*i* = 1, 2, 3) is the bending stiffness and the rotation angles of the flexure hinges, respectively; *F_leg_* is the force experienced by the leg; *δ_leg-y_* is the lift displacement of the leg; *m_a_* is the actuator mass; *m_i_* and *L_i_* (*i* = 0, 1, 2) are the masses and lengths of the links in the crank-slider mechanism, respectively; Δ*v_i_* and Δ*ω_i_* are the velocity changes and angular velocity changes of the links from the beginning to the end of the motion, respectively; similarly, Δ*v_leg_* and Δ*ω_leg_* are the velocity changes and angular velocity changes of the leg, respectively; and *J_i_* and *J_leg_* are the moments of inertia of the links and leg, respectively.

The resonance frequency of piezoelectric actuators is on the order of 1 kHz [30], while the operating frequency of the actuators in this paper is at a frequency below 100 Hz, so the actuator works quasi-statically. And the mass and damping can be neglected. In addition, the links in the transmission weigh approximately 1 mg, which can be regarded as massless links [31]. At the same time, the robot’s velocity changes very little during the quasi-static movement (1–10 Hz) [20]. Based on these analyses, simplifying Equation (1) and taking the derivative of t yields:(2)$$F_{a}\frac{\partial \delta_{a}}{\partial t}-k_{a}\delta_{a}\frac{\partial \delta_{a}}{\partial t}-k_{1}\phi_{1}\frac{\partial \phi_{1}}{\partial t}-k_{2}\phi_{2}\frac{\partial \phi_{2}}{\partial t}-k_{3}\phi_{3}\frac{\partial \phi_{3}}{\partial t}-F_{leg}\frac{\partial \delta_{leg-y}}{\partial t}=0$$

Dividing Equation (2) with *∂δa*/*∂t* yields:(3)$$F_{a}-k_{a}\delta_{a}-k_{1}\phi_{1}\frac{\partial \phi_{1}}{\partial \delta_{a}}-k_{2}\phi_{2}\frac{\partial \phi_{2}}{\partial \delta_{a}}-k_{3}\phi_{3}\frac{\partial \phi_{3}}{\partial \delta_{a}}-F_{leg}\frac{\partial \delta_{leg-y}}{\partial \delta_{a}}=0$$

According to Equation (3), the output force of the robot leg is determined by the output characteristics of the actuator, the mechanical properties of the flexure hinge, and the kinematic relationship of the crank-slider mechanism. The relationship between the output force and bending displacement of the beam end of the piezoelectric actuator is linear for the quasi-static motion [24], and the proportional coefficient is the stiffness *k_a_* of the actuator. Therefore, the output characteristic curves of *F_a_* and *δ_a_* can be obtained by linearly fitting the experimental results of the output force and output displacement with the least squares method. The experimental data and fitting results will be provided in Section 6.1. The kinematics of the crank-slider mechanism can be given by the following closed equation:(4)$$\left\{ \begin{array}{l} \delta_{leg-y}=L_{2}\cos\phi_{3}+L_{leg-x}\sin\phi_{3}+L_{leg-y}\cos\phi_{3}-L_{2}-L_{leg-y}\\ L_{1}\cos\phi_{1}=L_{1}+\delta_{a}-L_{2}\sin\phi_{3}\\ L_{1}\sin\phi_{1}=L_{2}-L_{2}\cos\phi_{3}\\ \phi_{2}=\phi_{1}+\phi_{3} \end{array}\right.$$

Substituting the solution of Equation (4) into Equation (3), the output force *F_leg_* of the leg can be given as:(5)$$\begin{array}{l} F_{leg}=\frac{1}{L_{1}\cos\phi_{1}\left[L_{leg-x}\cos\phi_{3}-\left(L_{2}+L_{leg-y}\right)\sin\phi_{3}\right]}\\ \{L_{1}L_{2}\cos(\phi_{1}+\phi_{3})\left[F_{a}-k_{a}(L_{1}\cos\phi_{1}+L_{2}\sin\phi_{3}-L_{1})\right]\\ -L_{2}k_{1}\phi_{1}\sin\phi_{3}-k_{2}\phi_{2}\left(L_{2}\sin\phi_{3}+L_{1}\cos\phi_{1}\right)-L_{1}k_{3}\phi_{3}\cos\phi_{1}\} \end{array}$$

Obviously, when the displacement of the robot leg is given and the stiffness of the flexure hinges is known, the leg output force is uniquely determined by the lengths of each link. By fitting the experimental data, it can be obtained that the actuator force *F_a_* = 195 mN and the actuator stiffness *k_a_* = 491 N/m (as we will describe in Section 5). Meanwhile, to reduce the impact of the robot sagging under load, the leg output displacement *δ_leg-y_* is specified as 1.5 mm. For the flexure hinges, we designed their lengths and widths to be 120 μm and 1.6 mm, respectively, for ease of manufacturing. Commercially available polyimide films for miniature robots are typically 20 μm, 30 μm, 40 μm, and 50 μm in thickness. Therefore, the stiffness of the flexure hinges can be calculated [25]. To sum up, with the link lengths (*L*_1_, *L*_2_, *L_leg-x_*, and *L_leg-y_*) as the design variable and the leg output force as the objective function, the optimization problem of the robot’s payload capacity is determined in this regard. We limit the length, width, and height dimensions of the whole robot to 5 cm × 5 cm × 3 cm, and, considering the assembly relationship between the various components of the robot, as well as the inherent geometrical constraints of the crank-slider mechanism, the constraints of this optimization problem can be given by the following equations:(6)$$\left\{ \begin{array}{l} 0\leq \phi_{1},\phi_{3}\leq \pi /2\\ 0\leq \phi_{1}+\phi_{3}\leq \pi /2\\ 3\;\mathrm{mm}\leq L_{1}\leq 8\;\mathrm{mm}\\ 0.5\;\mathrm{mm}\leq L_{2}\leq 5\;\mathrm{mm}\\ 1.5\;\mathrm{mm}\leq L_{leg-x}\leq 15\;\mathrm{mm}\\ 4\;\mathrm{mm}\leq L_{leg-y}\leq 15\;\mathrm{mm} \end{array}\right.$$

For flexure hinges with different thicknesses (20 μm, 30 μm, 40 μm, and 50 μm), the optimization problem was solved using the MATLAB Optimization Toolbox. The optimization results of the leg output force on different combination lengths [*L*_1_, *L*_2_, *L_leg-x_*, and *L_leg-y_*] of the links are shown in Figure 6a. The horizontal axis refers to the data points formed by different combinations of link lengths. The row vectors in Figure 6a represent the lengths (unit: mm) of various links corresponding to the maximum output force of the leg. It can be seen from Figure 6a that, for any thickness of the hinge, there exists an optimal linkage combination that maximizes the output force of the robot leg, and the output force converges with the number of iterations. It is clear that the robot leg has a maximum output force of 0.0236 N when the thickness of the flexure hinge is 20 μm. Based on the optimization results in Figure 6a, the relationship curve between the hinge thickness and the maximum leg force corresponding to the optimal linkage combination under this hinge thickness was plotted, as shown in Figure 6b. Obviously, the leg output force increases as the hinge thickness decreases. However, the robot body with 20 μm-thick flexure hinges has a sag of close to 2 mm in the robot locomotion test, so it can not produce effective movement. Therefore, we choose hinges with a thickness of 30 μm for the transmission. Correspondingly, the lengths of the links in the crank-slider mechanism are *L*_1_ = 8 mm, *L*_2_ = 1.2 mm, *L_leg-x_* = 7.4 mm, and *L_leg-y_* = 13.4 mm, respectively.

In order to express more intuitively the relationship between the leg output force and the lengths of the links, the constraint space of the four variables (*L*_1_, *L*_2_, *L_leg-x_*, and *L_leg-y_*) specified in Equation (6) is divided at 1 mm intervals, that is to say, *L*_1_ = [3 4… 8], *L*_2_ = [0.5 1.5… 4.5], *L_leg-x_* = [1.5 2.5… 14.5], and *L_leg-y_* = [4 5… 15]. *L*_1_, *L*_2_, *L_leg-x_*, and *L_leg-y_*, respectively, contain 6, 5, 14, and 12 elements. Then, four four-dimensional matrices containing 6 × 5 × 14 × 12 elements are generated using MATLAB’s ‘ndgrid’ function, and the corresponding elements of the four matrices at the same position constitute 5040 co-ordinates (i.e., the combination of lengths of each link). Taking two two-dimensional matrices M and N as an example, the corresponding elements of M and N at the same position refer to the two elements in the i-th row and j-th column of these two matrices. According to Equation (5), an image of the *F_leg_* = *f* (*L*_1_, *L*_2_, *L_leg-x_*, *L_leg-y_*) is plotted as shown in Figure 7. Obviously, different co-ordinate points correspond to different leg output forces, and there exists a co-ordinate point (combination of link lengths) that maximizes the leg output force.

## 5. Experiments

### 5.1. Force and Displacement Experiments of Piezoelectric Actuators

The piezoelectric actuator is driven by a sinusoidal drive signal with variable peak-to-peak voltage and frequency, which is generated by a signal generator and amplified by a voltage amplifier to drive the actuator. The block force data are measured and recorded by a six-axis force/torque sensor (ATI Industrial Automation, North Carolina, USA, Nano 17), and the bending displacement is measured and recorded by a laser displacement sensor (Keyence, LK-G30). The experimental setups for measuring the force and displacement of the actuator are shown in Figure 8a and b, respectively. The base of the piezoelectric actuator is fixed to a manual positioning stage by means of a customized fixture [12]. When measuring the block force, the end of the actuator is fixed on the sensor through a specially designed fixture. When measuring the bending displacement, the end of the actuator can bend freely. The position of the actuator is adjusted through the positioning stage. During the test, the upper limit of the drive voltage is set to 210 V to prevent the mechanical failure of the piezoelectric ceramics.

### 5.2. The Leg’s Quasi-Static Force Experiments

In order to test the output capability of a single leg of the robot during quasi-static motion and to verify the accuracy of the powertrain model in Section 2, the experimental setup for leg force/displacement measurements is shown in Figure 9. The robot is fixed to a three-axis positioning stage by a customized clamping fixture, and the distance between the tip of the robot’s leg and the force sensor (i.e., the displacement of the leg *δ_leg-y_*) is adjusted by changing the height of the positioning stage. The positioning accuracy of the positioning stage is 0.02 mm. During the experiment, the distance between the leg and the positioning stage is changed at intervals of 0.2 mm until the leg cannot contact the sensor under the actuator drive.

### 5.3. Dynamic Model Identification Experiments of the Powertrain

When the drive frequency of the robot’s powertrain system is close to or even exceeds its resonance frequency, the robot has a higher speed of movement and a lower cost of transport [32]. Therefore, the model identification of the powertrain system is necessary. To characterize the frequency response of the powertrain at different drive frequencies, the eight DOFs of the four legs of the robot are driven with sinusoidal signals independently and the eight outputs (four lifts and four swings) of the legs are measured separately. The experimental setup is shown in Figure 10. The laser displacement sensor measures and records the output displacement of robot legs at different drive frequencies. In order to ensure that the laser is always projected onto the end of the leg during its movement, a round cardstock with a certain area is glued to the end of the leg.

### 5.4. Locomotion Test of the Robot

Before conducting the locomotion performance test of the robot, the control methods of the eight actuators of the robot need to be clarified first. The layout diagram of the actuators below the circuit board is shown in Figure 11a. The arrows represent the bending direction when eight actuators share the same drive signal. The piezoelectric actuator has three pads connected to bias voltage, drive voltage, and ground, respectively. To make the robot move forward in a trot gait, the drive signals of the eight actuators are shown in Figure 11b. There is a 90-degree phase difference between the three drive voltages. The subscript numbers represent the numbering of each actuator. For example, actuators 1, 3, 5, and 7 share the same drive signal. Numbers Ⅰ–Ⅴ represent five feature points within one cycle of the drive signal. The positions of the robot leg corresponding to these five feature points are shown in Figure 11c. The arrows in Figure 11c represent the direction of motion of the robot legs. The “N” point represents the neutral point where the leg does not move. Connecting these five positions forms an approximate elliptical motion trajectory for the robot leg. The movement direction of the two diagonal legs is completely the same, while the two legs on the same side are completely opposite. As a result, the robot’s footfall patterns are shown in Figure 11d.

Under the drive signals shown in Figure 11b, the robot can achieve forward motion. The performance of the forward movement was tested. The experimental setup consists of a pad with scales as a walking surface and a high-speed camera located on the side of the robot, as shown in Figure 12. The distance between the two scale lines on the pad is 1 cm. By combining the frame per second (FPS) of the high-speed camera with the frame difference of the robot at different positions, the movement time of the robot can be determined. The movement distance of the robot can be obtained by the scale on the pad, so the speed of the robot can be calculated.

## 6. Results and Discussion

### 6.1. Forces and Displacements of Piezoelectric Actuators

Three different actuator versions were measured to compare their block forces and displacements, and the experimental results are shown in Figure 13a and b, respectively. The drive voltage applied to the actuator is a sinusoidal voltage, and the displacement and block force of the actuator are also sinusoidal signals. For convenience, the voltage, displacement, and block force in Figure 13a,b refer to the peak-to-peak value of sinusoidal signals. The ‘AC’ in Figure 13 refers to an actuator with an alumina base and CF attachment, the ‘AF’ refers to an actuator with an alumina base and FR-4 attachment, and the ‘FG’ refers to an actuator with an FR-4 base and GF attachment. Each error bar is the standard deviation of the 15 pieces of data obtained from five actuators. The ‘AC’ actuator used in this paper with an alumina base and CF attachment ‘bridging’ the PZT–alumina interface has a greater output force due to its stronger base and attachment, which effectively avoids the parasitic bending that occurs at the interface. The maximum displacements of the three versions of the actuator do not differ much.

As shown in Figure 8a, we change the distance between the end of the actuator and the force sensor by adjusting the height of the positioning stage, using this distance as the bending displacement of the actuator, and measure the output force *F*_out_ corresponding to this bending displacement *δ_a_* driven at a 210 V DC voltage. The results are shown in Figure 13c, the slope of the curve is the stiffness *k_a_* of the actuator, and the force *F_a_* is the output force generated when the end displacement of the actuator is 0, that is, *k_a_* = 491 N/m, and *F_a_* = 195 mN.

### 6.2. The Leg’s Quasi-Static Force Results

The experimental results of the leg’s quasi-static forces are shown in Figure 14. It can be seen that the output force of each leg is sufficient to carry half of the robot’s weight, which is a necessary condition for the robot to successfully move in a trot gait. The simulation results in Figure 14 are obtained from the powertrain model in Section 4. According to the analysis in Section 4, the thickness of the flexure hinge is selected as 30 μm, so the bending stiffness of the hinge *k_i_* was determined. The corresponding optimal linkage combination (*L*_1_, *L*_2_, *L_leg-x_*, and *L_leg-y_*) has also been determined. The piezoelectric force *F_a_* of the actuator and the stiffness *k_a_* are also analyzed in Section 6.1. Therefore, except for the parameters of leg force and displacement, all other parameters in Equation (5) are known, and the output force of the robot leg is only determined by the leg displacement. Therefore, the relation of leg force *F_leg_* versus lift displacement *δ_leg-y_* can be obtained through MATLAB simulation. The experimental results of the robot’s leg force are lower than the simulation results, because the exoskeleton of the robot is not rigid and the flexure hinges will buckle when subjected to load, and the bending displacement of the actuator is already greater than the theoretical displacement.

### 6.3. Dynamic Model Identification Results of Powertrain

Based on the measurement results of leg displacements, we plotted Bode plots for each DOF of the robot‘s leg, where the amplitude was normalized to the maximum displacement of the actuator and the phase offset was obtained by comparing the sinusoidal input signal with the out displacement of the leg. Taking the powertrain corresponding to the front left leg of the robot as an example, its Bode plot is shown in Figure 15. Due to the design parameters of the other three powertrains being the same as that of the left front leg, their frequency response is theoretically the same. The robot’s powertrain is a second-order oscillation system whose amplitude–frequency and phase–frequency characteristics can be expressed as:(7)$$\begin{array}{l} L\left(\omega \right)=20\mathrm{lg}\frac{k}{\sqrt{{\left(1-\frac{\omega^{2}}{\omega_{n}^{2}}\right)}^{2}+{\left(2\xi \frac{\omega }{\omega_{n}}\right)}^{2}}}\\ \varphi (\omega )=-\arctan\frac{2\xi \frac{\omega }{\omega_{n}}}{1-{\left(\frac{\omega }{\omega_{n}}\right)}^{2}} \end{array}$$
where *ω* is the frequency of the drive signal, *ω_n_* is the natural frequency, *k* is the amplification coefficient, and *ξ* is the damping ratio. Based on this second-order oscillation model, the experimental results are fitted by MATLAB’s fittype and fit functions. The damping ratios for the swing and lift DOFs of the front left leg are 0.0685 and 0.0866, respectively; the natural frequencies are 61.88 Hz and 72.17 Hz, respectively; and the amplification coefficients are 3.499 and 3.428, respectively. The results indicate that the front left powertrain is under-damped and there is a 90° phase offset between the leg output and input signals when the drive frequency is close to the natural frequency.

### 6.4. The Locomotion Performance of the Robot

The speeds of the robot without any payload at different drive frequencies were measured. The experimental results are shown in Figure 16a. Since the swing amplitude of the leg directly affects the stride length of the robot, the frequency of the horizontal axis co-ordinate in Figure 16a is the drive frequency of the swing actuator. In order to obtain the required leg trajectories and walking route, the input signal is adjusted with a phase shift based on the second-order system fit in Section 3. At the same time, the speeds of the robot with different payloads were also measured; the results are shown in Figure 16b. The data points represented by black solid circles are the experimental results of the robot’s powertrain being driven in a resonant state, where the drive frequency of the lift actuators and the swing actuators are 72 Hz and 61 Hz, respectively. The data points represented by the red triangle are experimental results with a quasi-static drive frequency of 10 Hz.

The results shown in Figure 16a indicate that the robot has a maximum speed of 48.66 cm/s when the drive frequency of the lift DOF is 70 Hz, which is 8.8 times the quasi-static speed (the drive frequency is 10 Hz). This is consistent with the frequency response results of the powertrain shown in Figure 15. When the drive frequency of the swing DOF is 72 Hz, the swing amplitude of the leg is the maximum. As shown in Figure 16b, the robot has a maximum payload capacity of 5.5 g when the powertrains are driven in a resonant state, which is 3.05 times the robot’s mass. The load capacity of the robot under dynamic motion is 175% higher than that under quasi-static motion. The video of the robot moving forward at maximum speed and moving with a 5.5 g payload is available in Appendix A. Figure 17 shows representative frames captured by a high-speed camera during the locomotion of the miniature robots.

To demonstrate the improvement of the motion performance of the miniature robot designed in this paper more intuitively, our robot SMR-O was compared with several reported classical miniature robots. The comparison results are shown in Table 1. Compared to HAMR, our robot has a faster speed and greater payload capacity. Compared to DASH and MinRAR, our robot has a much lighter mass than theirs. In addition, our robot has a much faster speed than RoACH at similar masses and lengths. Moreover, the legs, exoskeletons, and hip joints of the robot in this paper can be integrated into a single laminate for monolithic manufacturing. The monolithic integrated component is precisely folded after being released to form the robot’s frame and movable structure [29].

## 7. Conclusions

This paper presented a miniature quadrupedal piezoelectric robot with a mass of 1.8 g and a body length of 4.6 cm. A new manufacturing method for piezoelectric actuators is proposed, which uses discrete patterned PZT pieces during material stack to avoid the cutting of PZT by a high-power laser. The base of the actuator is alumina, and the PZT–alumina interface is bridged with carbon fiber. Compared to the actuator using FR-4 as the base and attachment, the output force of the actuator in this paper has increased by 32%. The lift powertrain of the miniature robot is modeled. Based on this model, the link lengths of the powertrain were optimized, and the maximum output force of each leg can reach 26 mN, which is 138% of the force required for a robot to successfully walk in a trot gait. The frequency response of the powertrain was measured and the results were fitted with the second-order system to adjust the drive signal of the miniature robot to obtain the required leg trajectory. When the powertrain of the robot is driven to a resonant state, the maximum speed of the robot without load can reach 48.66 cm/s, and the payload capacity can reach 5.5 g. The locomotion performance is greatly improved compared with the speed of 5.32 cm/s and the payload capacity of 2.5 g at a drive frequency of 10 Hz.

## Figures and Tables

**Figure 1 biomimetics-09-00226-f001:**
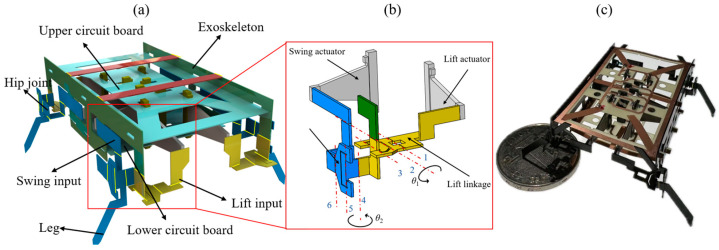
(**a**) A CAD model of the miniature robot. (**b**) A schematic diagram of the powertrain. (**c**) The prototype of the miniature robot contrasted with a coin.

**Figure 2 biomimetics-09-00226-f002:**
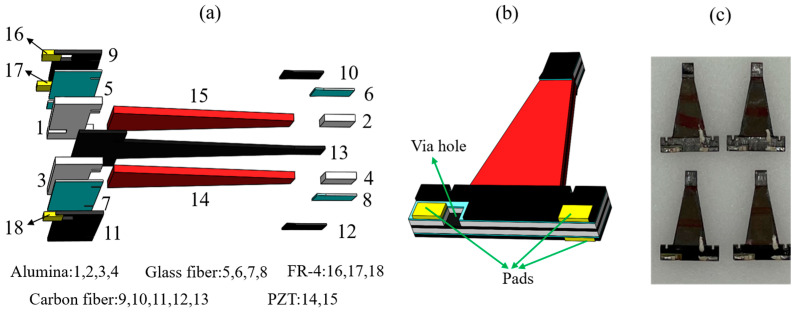
(**a**) Explosion view of the actuator. (**b**) The CAD model of the actuator. (**c**) The physical picture of the actuators.

**Figure 3 biomimetics-09-00226-f003:**
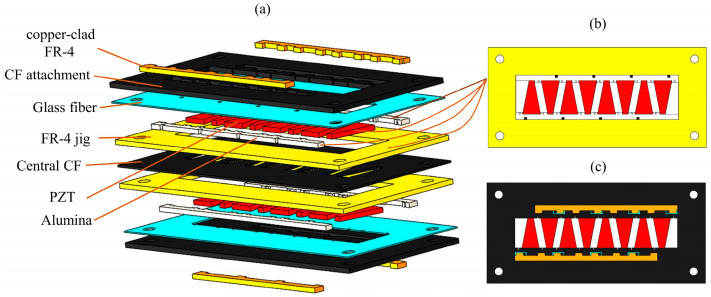
(**a**) The stacking order of the actuator components. (**b**) The positional relationship of the alumina, PZT, and FR-4 jig in the same plane. (**c**) The top view schematic diagram of the stacked laminate.

**Figure 4 biomimetics-09-00226-f004:**
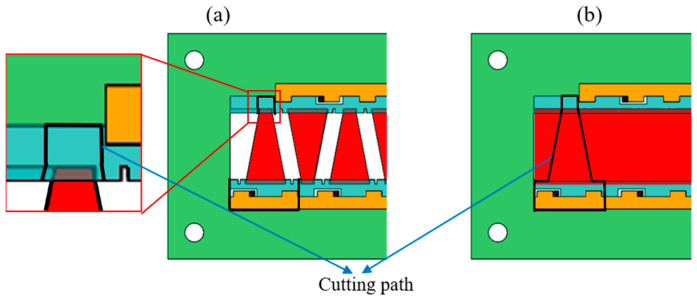
(**a**) The stacked laminate using discrete PZT pieces and its cutting path. (**b**) The stacked laminate using a whole piece of PZT and its cutting path.

**Figure 5 biomimetics-09-00226-f005:**
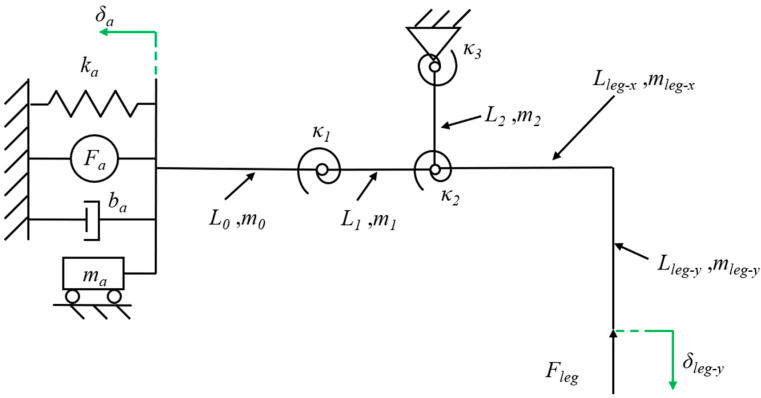
The simplified model drawing of the lift powertrain.

**Figure 6 biomimetics-09-00226-f006:**
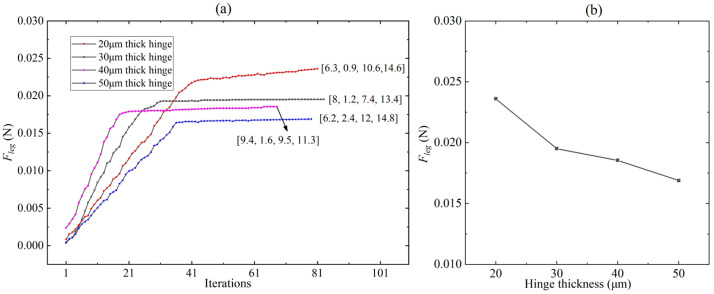
(**a**) The optimization results of the leg output force on different combination lengths [*L*_1_, *L*_2_, *L_leg-x_*, and *L_leg-y_*] of the links for flexure hinges of different thicknesses. (**b**) Relationship between different hinge thicknesses and the corresponding maximum leg output forces.

**Figure 7 biomimetics-09-00226-f007:**
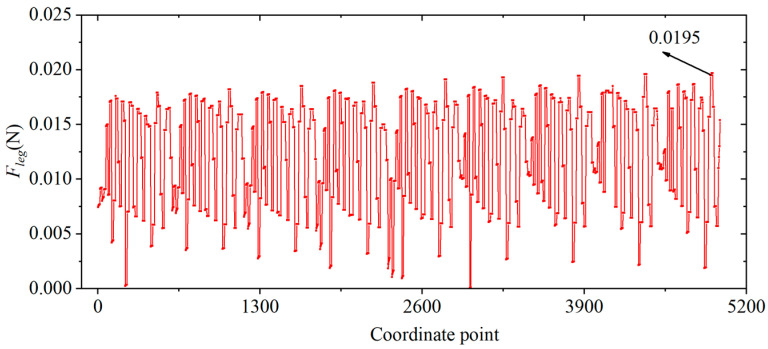
The relationship between the output force of the legs and the co-ordinate points formed by different combinations of link lengths.

**Figure 8 biomimetics-09-00226-f008:**
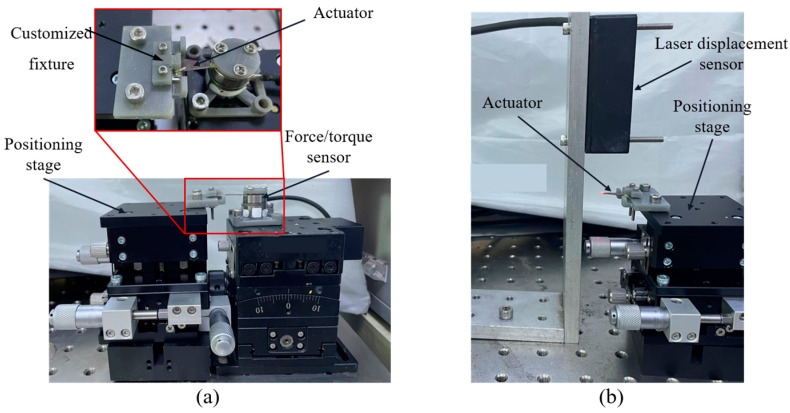
(**a**) Experimental setup for measuring actuator forces. (**b**) Experimental setup for measuring actuator displacements.

**Figure 9 biomimetics-09-00226-f009:**
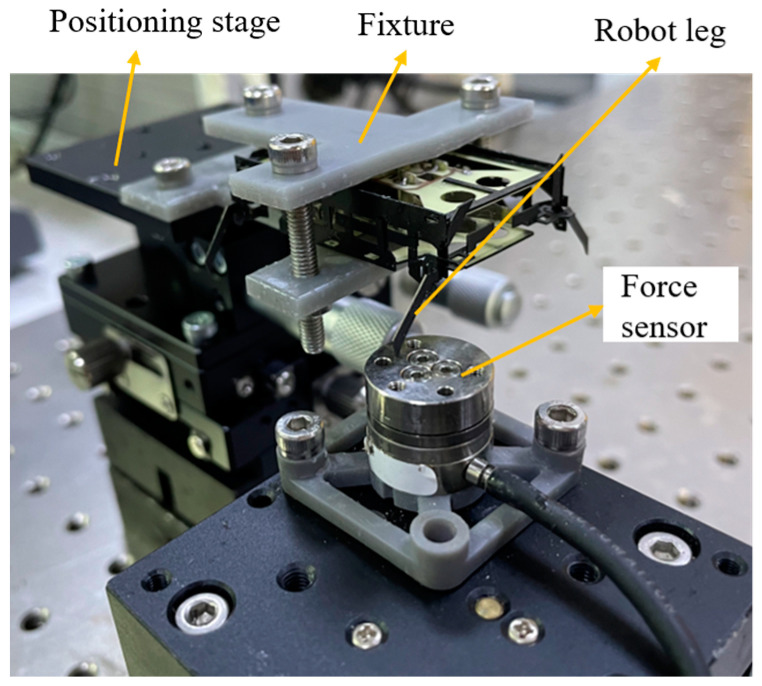
The experimental setup for the force measurement of a leg of the robot.

**Figure 10 biomimetics-09-00226-f010:**
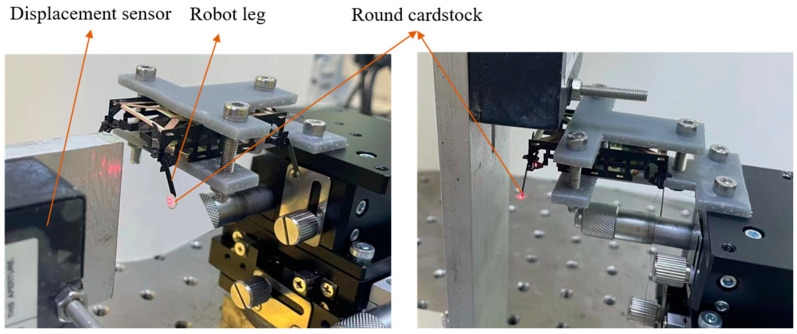
Experimental setup for frequency response of the swing (**left**) and lift (**right**) DOFs of the leg.

**Figure 11 biomimetics-09-00226-f011:**
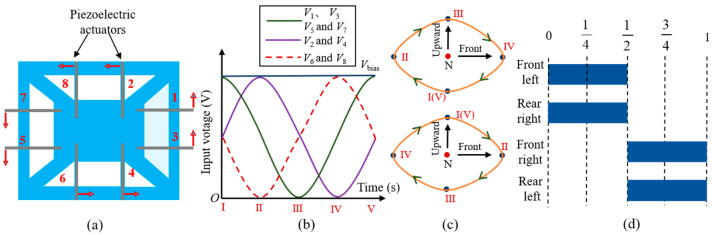
(**a**) Layout diagram of the actuators below the circuit board. (**b**) The drive signals of eight actuators for the robot to move forward in a trot gait. (**c**) The motion trajectory of two robot legs on the same side under the drive signal of trot gait. (**d**) Footfall patterns of the trot gait.

**Figure 12 biomimetics-09-00226-f012:**
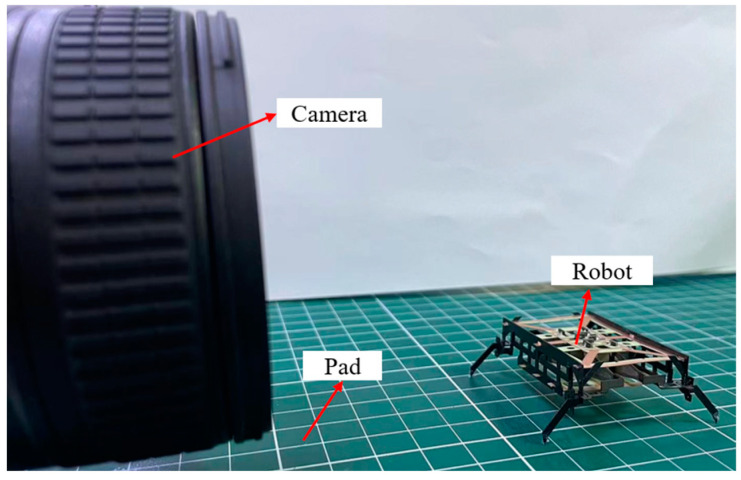
Experimental setup for robot locomotion test.

**Figure 13 biomimetics-09-00226-f013:**
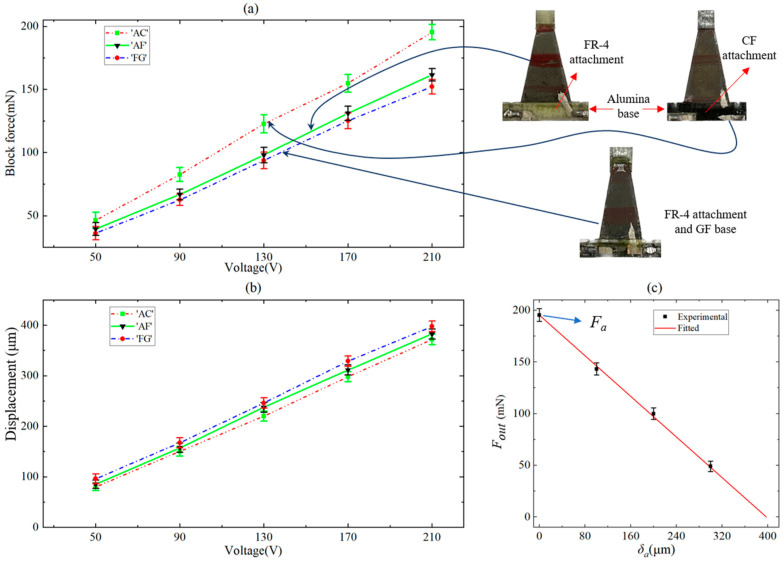
(**a**) Experimental results of the block force (peak-to-peak value) of three different actuator versions. (**b**) Experimental results of the displacement (peak-to-peak value) of three different actuator versions. (**c**) Force–displacement curves for ‘AC’ actuators driven at 210 V. Each error bar is the standard deviation acquired from five actuators with the same design parameters.

**Figure 14 biomimetics-09-00226-f014:**
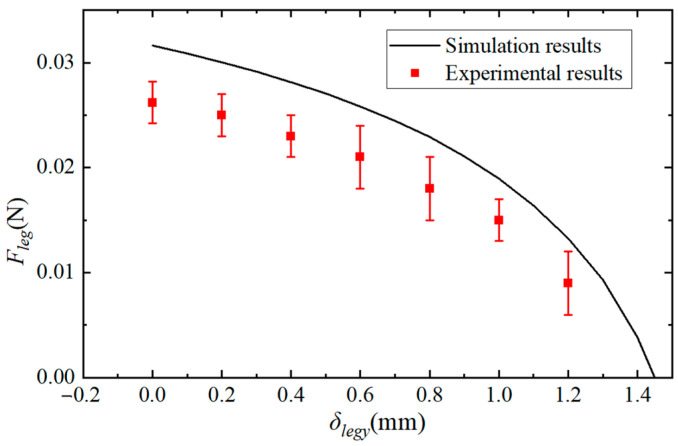
The experimental and simulation results of the leg force at different displacements. Each error bar of the experimental results is the standard deviation of the five repeated experiments.

**Figure 15 biomimetics-09-00226-f015:**
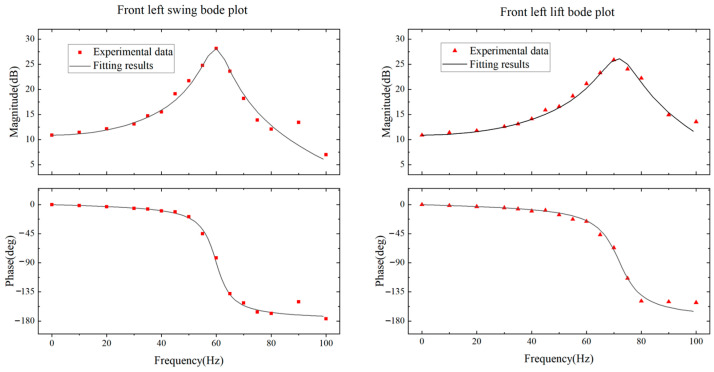
Frequency responses and second-order oscillation model fitting of the powertrain of the front left leg.

**Figure 16 biomimetics-09-00226-f016:**
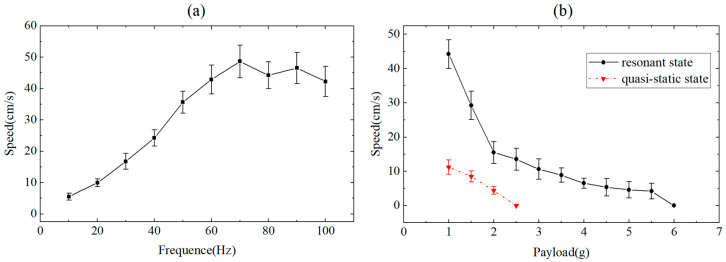
(**a**) The speed of the robot with no payload under different drive frequencies. (**b**) The speed of the robot with different payloads. Each error bar is the standard deviation of five repeated experiments.

**Figure 17 biomimetics-09-00226-f017:**
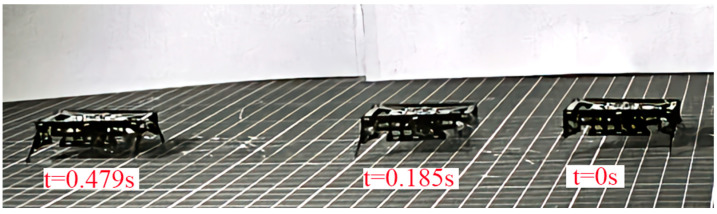
Representative frames captured by camera when the miniature robot reaches a speed of 48.66 cm/s.

**Table 1 biomimetics-09-00226-t001:** Comparison of current autonomous miniature robots.

Robots	Length (cm)	Mass (g)	Maximum Speed (cm/s)	Highlights	Refs
SMR-O	4.6	1.8	48.66	Greater payload capacity of 5.5 g compared to HAMR; exoskeletons, legs, and hip joints monolithically integrated and manufactured.	This paper
HAMR-VP	4.4	1.27	37	Improved manufacturing and assembly speed of robots through the pop-up process. Increased payload capacity to 1.35 g compared to previous versions of robots.	[14]
DASH	10	16.2	150	The highest speed among miniature robots.	[17]
MinRAR	5.5	16	52	A fast speed driven at the resonance frequency.	[26]
RoACH	3	2.4	3	The first robot to use smart composite microstructure technology.	[9]

## Data Availability

The data are available from the corresponding author upon reasonable request.

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
