# Peer review of "Development and Improvement of a Piezoelectrically Driven Miniature Robot"

_biomimetics, 2024, doi:10.3390/biomimetics9040226_

Round 1

Reviewer 1 Report (Previous Reviewer 1)

Comments and Suggestions for Authors

Dear editor,

this work details a 1.8g, 4.6cm long piezoelectric quadruped robot based on HAMR designs with innovative leg mechanisms and manufacturing techniques. It highlights the robot's optimized powertrain for improved force and speed, achieving a maximum speed of 48.66 cm/s and a payload capacity of 5.5g, significantly surpassing its weight.

In the revised version, authors improved the comparison with other robots and they attributed the mechanical optimization to insect joints inspiration. The submission still represents an incremental improvement on existing HAMR designs, lacking significant novelty, although they carried out a very exhaustive analysis of their robot. I recommend its publication in present form.

Best

Author Response

  Thank you very much for taking the time to review this manuscript. Please find the detailed responses below and the corresponding revisions highlighted in the revised files. 

Point-by-point response to Comments and Suggestions for Authors

Comment 1: [

this work details a 1.8g, 4.6cm long piezoelectric quadruped robot based on HAMR designs with innovative leg mechanisms and manufacturing techniques. It highlights the robot's optimized powertrain for improved force and speed, achieving a maximum speed of 48.66 cm/s and a payload capacity of 5.5g, significantly surpassing its weight.

 In the revised version, authors improved the comparison with other robots and they attributed the mechanical optimization to insect joints inspiration. The submission still represents an incremental improvement on existing HAMR designs, lacking significant novelty, although they carried out a very exhaustive analysis of their robot. I recommend its publication in present form.

]

Response 1: Thank you very much for pointing this out and recommendation for publication in the current form. The innovation of this manuscript lies in the use of a new spatial parallel mechanism as the transmission mechanism of the robot. The use of the new mechanism allows the exoskeleton, hip joint, and leg to be integrated into a laminate, and the mechanical structure of the robot can be obtained through one stacking and one release. As reported in this manuscript, we have optimized the transmission of the robot and the fabrication process of the actuator, greatly improving its motion performance. Considering that we have developed the robot SMR-O in this manuscript, which has fast speed and large payload, in future work, we will use SMR-O as an experimental platform, integrating control circuits, cameras, gyroscopes, sensors, miniature batteries, etc., making SMR-O an autonomous and untethered robot to perform reconnaissance, detection, and other tasks.

Reviewer 2 Report (Previous Reviewer 2)

Comments and Suggestions for Authors

The revised manuscript is acceptable.

Author Response

Thank you very much for taking the time to review this manuscript. 

Point-by-point response to Comments and Suggestions for Authors

Comments 1: [

The revised manuscript is acceptable.

]

Response 1: Thank you very much for your approval of the revised manuscript.

Reviewer 3 Report (Previous Reviewer 3)

Comments and Suggestions for Authors

no

Comments on the Quality of English Language

no

Author Response

Thank you very much for taking the time to review this manuscript.

Point-by-point response to Comments and Suggestions for Authors

Comments and Suggestions for Authors: No

Response: Thank you for reviewing our manuscript. 

This manuscript is a resubmission of an earlier submission. The following is a list of the peer review reports and author responses from that submission.

Round 1

Reviewer 1 Report

Comments and Suggestions for Authors

Translator        

Translator        

Dear authors,

After a thorough analysis, I regret to inform you that I recommend rejecting this submission for the following reasons:
1) Scope misalignment: The manuscript primarily discusses mechanical and manufacturing optimizations of a quadrupedal piezoelectric robot, deviating from the journal's biomimetic theme. The work centering instead on engineering enhancements.
2) Lack of comparative analysis: It omits a crucial comparison with current autonomous miniature robots, including those based on the HAMR design. This comparison is essential for contextualizing the biomimetic relevance of the proposed improvements.

The submission represents an incremental improvement on existing HAMR designs without sufficient biomimetic innovation. I suggest the authors consider a publication venue more focused on mechanical engineering or robotics.

Best regards

Reviewer 2 Report

Comments and Suggestions for Authors

In this work, the authors prepared a miniature quadrupedal piezoelectric
robot with a mass of 1.8 g and a body length of 4.6 cm. The robot adopts a
novel spatial parallel mechanism as its transmission. A new  manufacturing method for piezoelectric actuators was proposed. The maximum output force of each leg can reach 26 mN under the design parameters. The authors reported that the maximum speed of the robot without load reached 48.66 cm/s and the payload capacity can reach 5.5 g near the powertrain resonance. In general, the work is very interesting. However, some issues should be addressed as follows:

1.    The structure of the manuscript should be re-organized. Since the experimental section was merged with the result section. Therefore, it is difficult for the readers to follow the story of the manuscript.

2.    The control methods and the algorithm to combine the phase of the 4 piezoelectric actuators should be described in detail.

3.    Locomotion test: simulation experiments should be performed and compared with the actual experiments.

4.    The quality of the image in Figure 15 should be improved.

Reviewer 3 Report

Comments and Suggestions for Authors

It is a practical and interesting work on the leg design of a micro robot. However, there are unclear and improper contents as well as uncertain contribution which the Authors claims the contributions as below.

1)     Manufacturing method: instead of using a whole PZT, a combination of several pieces of materials with special shape and manufacturing process is proposed.

Question and deficiency) It seems very practical by showing the influence to leg’s performance in experiment but lack of in-depth theoretical approach or even a design and manufacturing rule. That is, the proposed design is only a feasible way. Also, for example, ”The central carbon fiber is sandwiched between two layer of PZT”, is actually a well-known bimorph design in piezoelectric actuator.

2)     Optimization of leg’s length.

“the output force of the robot leg is determined by the output characteristics of the actuator, the mechanical properties of the flexure hinge, and the kinematic relationship of the crank-slider mechanism. The output characteristics of the actuator can be fitted by the experimental results”.

Question and deficiency)

i)    How to derive eq.(3)?

ii)   What are the procedures on “The output characteristics of the actuator can be fitted by the experimental results”?  It is very vague if the Authors attempt to create a contribution on design procedure.

iii) Also, the optimization is not solid and convergent by several trials on the chosen numerous hinge thickness. 

3)     Motion speed: “By driving the powertrain of the robot to a resonant state, the motion performance has been greatly improved.”

Question and deficiency) It is well known that driving piezoelectric actuator at resonance frequency (with desired modal shape) would achieve higher output. 

Other inaccuracies or unclear description are:

(1)   Need further description in Figure 7. What are the corresponding elements of the “four matrices”?

(2)   How to have the “Fitted” data in Figure 8?  What are the obtained signals of force and displacement while applying AC signals to the actuator in Figures 8 and 10?  How to find the relation of force versus displacement in simulation in Figure 10?

(3)   In eq.(7), it should be “k is the amplification coefficient and x is the damping coefficient”. What is the value of k?

(4) In case of importance of novel design shown in Figure 1~4, further in-depth analysis and methodology need to systematically described.

Comments on the Quality of English Language

readable